# Ultrasound for the Detection of Inflammatory Abdominal Aortic Aneurysms: A Case and Validation Series

**DOI:** 10.3390/diagnostics13101669

**Published:** 2023-05-09

**Authors:** Berend G. C. Slijkhuis, David J. Liesker, Sherilyn A. C. Konter, Annet Possel-Nicolai, Reinoud P. H. Bokkers, Niek H. J. Prakken, Elisabeth Brouwer, Riemer H. J. A. Slart, Arie M. van Roon, Ben R. Saleem, Douwe J. Mulder

**Affiliations:** 1Department of Internal Medicine—Vascular Medicine, University Medical Center Groningen, University of Groningen, 9712 CP Groningen, The Netherlands; 2Department of Surgery—Vascular Surgery, University Medical Center Groningen, University of Groningen, 9712 CP Groningen, The Netherlands; 3Department of Radiology, University Medical Center Groningen, University of Groningen, 9712 CP Groningen, The Netherlands; 4Department of Rheumatology and Clinical Immunology, University Medical Center Groningen, University of Groningen, 9712 CP Groningen, The Netherlands; 5Department of Nuclear Medicine and Molecular Imaging, University Medical Center Groningen, University of Groningen, 9712 CP Groningen, The Netherlands; 6Biomedical Photonic Imaging Group, University of Twente, 7522 NB Enschede, The Netherlands

**Keywords:** inflammatory abdominal aortic aneurysm, ultrasound, periaortitis, IgG4-related disease, aortitis, chronic periaortitis, retroperitoneal fibrosis

## Abstract

Inflammatory abdominal aortic aneurysms (iAAA) are a form of noninfectious aortitis in patients with abdominal aortic aneurysms (AAA). Ultrasound could help to detect iAAA early. This retrospective observational study assessed the potential of using ultrasound to detect iAAA in a case series of iAAA patients, and the diagnostic value of ultrasound to detect iAAA in consecutive patients in a follow-up for AAA, referred to as a feasibility study. In both studies, diagnosis of iAAA was based on a cuff surrounding the aneurysm using CT (golden standard). The case series included 13 patients (age 64 (61; 72) years; 100% male). The feasibility study included 157 patients (age 75 (67; 80) years; 84% male). In the case series, all iAAA patients showed a cuff surrounding the aortic wall on ultrasound. In the feasibility study with AAA patients, ultrasound yielded no cuff in 147 (93.6%; CT negative in all cases), a typic cuff in 8 (5.1%; CT positive in all cases), and an inconclusive cuff in 2 (1.3%; CT negative in both cases) patients. Sensitivity and specificity were 100% and 98.7%, respectively. This study indicates that iAAA can be identified with ultrasound, and safely ruled out. In positive ultrasound cases, additional CT imaging might still be warranted.

## 1. Introduction

The prevalence of aneurysms of the abdominal aorta (AAA) has been steadily increasing in the last few decades, with current prevalence rates ranging from 1.3 to 8.9% in males and 1.0 to 2.2% in females [1,2]. In 5–10% of AAA, the disease progresses to an inflammatory abdominal aortic aneurysm (iAAA), which is part of a disease cluster called Chronic Periaortitis (CP), now also called noninfectious aortitis; the other diseases are idiopathic retroperitoneal fibrosis (IRF), and perianeurysmal retroperitoneal fibrosis (PRF or RPF). Of these diseases, perianeurysmal RPF is thought to be a progression of iAAA [3]. CP as a whole has an unclear etiology, some have speculated that extended response to atherosclerosis might be the case, although concerning IRF, it also occurs without atherosclerosis [4]. More recently, IgG4-related disease (IgG4-RD), has been linked to iAAA, being present in up to 57% of iAAA cases [5], having prompted some to classify iAAA as either IgG4-iAAA or non-IgG4-iAAA [6]. However, its pathophysiology remains unclear. Long-term use of corticoids and other immunosuppressive drugs reduces the extent of the iAAA, which also supports the idea of an immune-mediated origin of the disease [1,3]. Furthermore, iAAA is associated with increased morbidity [2] and potentially higher mortality rates, emphasizing the need for early recognition, thorough evaluation, and an early start of immunosuppressive treatment in most cases.

Inflammatory abdominal aortic aneurysms vary in type and origin and can present with varying degrees of complexity, ranging from exhibiting increased vessel wall thickness, to an increase in perianeurysmal tissue and RPF and adhesions of intra-abdominal tissue [7]. Left untreated, it can lead to severe complications such as progressive ureteral obstruction, hydronephrosis, and even irreversible renal dysfunction. Furthermore, peri-aortic inflammation can lead to hydrocele and venous problems such as thrombosis (in the case of compression of the gonadal vessels, the vena cava inferior, and iliac vein involvement, respectively) [1]. While this disease is often overlooked, it is important to detect, and start medical intervention in time, preventing irreversible complications.

Diagnosing iAAA typically involves a combination based on the clinical presentation, laboratory findings, and imaging, with computed tomography (CT) being the routine initial choice. Alternative diagnostic tools, such as 18F-fluorodeoxyglucose positron emission tomography (18F-FDG-PET) and MRI, may also be used. Presently, only one case–control study indicated that ultrasound imaging could play a role in the early diagnostic process [8], particularly due to its availability, low cost, and routine use in AAA follow-up [9].

A retrospective study was conducted to further investigate the potential of ultrasound imaging in the diagnosis of iAAA. The study aimed to assess the value of US imaging in two cohorts of patients, including a unique case series of patients with definite iAAA, and a second group of consecutive patients who were on AAA follow-up to investigate the diagnostic value of ultrasound in the routine follow-up of AAA (feasibility study). Ultrasound characteristics were identified and reported, and findings were validated using CT imaging. 

## 2. Materials and Methods

### 2.1. Patient Selection and Data Collection

In the first study, a case series was retrospectively assembled, consisting of 13 patients who visited our hospital between 2013 and 2020, without prior surgical intervention of the abdominal aorta and who were diagnosed with iAAA according to the guidelines of the European Society of Vascular Surgery (ESVS), and who had previous CT imaging results available [10]. Additionally, they underwent ultrasound imaging (Appendix A). These ultrasound images were subsequently assessed for potential abnormalities, as described below.

In the second study, we subsequently validated our potential identifiers for iAAA from the initial case series. We assessed the feasibility and diagnostic accuracy in all patients in our hospital from 2017 until 2021, who were in standard ultrasound follow-up for AAA. The diagnosis of AAA was made in accordance with the ESVS guidelines [10]. Only patients without prior surgical intervention of the abdominal aorta were included. Initially, 193 patients were identified as potentially eligible. After the subsequent exclusion of patients without previous available CT images, a group of 157 patients was finally selected. (Appendix A, shows ultrasound imaging for all positive cases).

For ultrasound imaging in the case series, the maximal anteroposterior AAA diameter, the presence of a cuff, and its maximal thickness were measured with brightness mode (B-mode) sonography, on a Siemens Acuson S2000 HELX TC (Siemens Medical Solutions, Mountain View, CA, USA) with a 6 MHz curved array transducer (Siemens Acuson 6C1 HD), using the 4 MHz harmonics mode (2 MHz transmission, and 4 MHz imaging frequency). Due to the retrospective nature of this study, no Doppler imaging was recorded, as the ultrasound imaging was only made to evaluate the AAA diameter.

The measurements used were performed in the transverse plane at the level of the maximum abdominal aortic diameter. The following ultrasound measurements were performed (Figure 1):The maximum anteroposterior diameter of the aorta including the hypo-echogenic layer outside the calcified layer.The anteroposterior diameter of the aorta excluding the hypo-echogenic layer (up to the calcified layer)The thickness of the hypo-echogenic layer at the anterior site of the aneurysm.

For the AAA group in the feasibility study, the existence of a hypoechogenic cuff was only recorded as a dichotomic variable.

Ultrasound imaging was performed by experienced vascular technicians and measurements were prospectively repeated by two independent assessors, blinded for clinical information. The first assessor was a student (case series: DL; AAA group: SK) trained in recognizing iAAA on ultrasounds, and the second assessor was a medical doctor (UM) with ample experience in evaluating ultrasound imaging of the vascular wall; the student made the initial selection of possible cases, after which the medical doctor gave the final evaluation of the ultrasound imaging in the cases flagged by the student, independent of, and blinded for, the final diagnosis. 

The ESVS criteria for iAAA included the presence of an unusually thickened aneurysm wall, shiny white peri-aneurysmal and retroperitoneal fibrosis, and dense adhesions of adjacent intra-abdominal structures [10]. Furthermore, the following conditions potentially mimicking iAAA were excluded: malignancies, hemorrhage, and a displaced duodenum on CT imaging, in either the feasibility or AAA diagnostic study. 

CT imaging was performed on multiple CT scanners in routine care, utilizing and images were generated with a slice thickness of 0.75 mm, tube voltage of 80 kV, a reference exposure of 183 mAs, and the iodine contrast agent Iomeron at 350 mg/mL. In line with the ESVS guidelines, CT imaging was considered positive for iAAA when “the mantle sign” also known as the cuff was present [10]. Currently, there is no clear consensus on how to measure the “diameter” of an inflammatory aneurysm, therefore we dichotomized the outcome of the CT as mantle sign present or not present. No thickened wall was measured because of the varying CT scanners and protocols used and the study’s retrospective nature. Absence of CT imaging was defined as those who lacked CT imaging available within one year of the assessed ultrasound imaging.

IgG4-RD, which is associated with the occurrence of iAAA, was defined as probable IgG4-RD when a mantle sign was present on CT imaging in which mainly the posterior aorta showed change and/or retroperitoneal fibrosis, as well as one of the following: elevated serum IgG4 (higher than 2 g/L) [11], or the presence of diseases associated with IgG4-RD. IgG4-RD was considered definitive when a biopsy was positive for IgG4. 

In both groups, demographic characteristics, comorbidities, symptoms, laboratory findings, microbiology results, pathology reports, imaging studies, treatment (both medical and surgical), and outcomes were recorded for patients with a confirmed diagnosis (in accordance with the ESVS guidelines) of iAAA.

Ethics approval was granted by the University of Groningen Medical Center Human Research Ethics Committee (deemed exemption for review according to the “Wet Medisch Onderzoek” (WMO)).

### 2.2. Statistical Analysis

Statistical analysis was performed with SPSS version 28 (IBM Corp., Released 2021, Armonk, NY, USA). Results are given as a number with according percentages for categorical data and skewed distributed data were reported as median (interquartile range (IQR): 25th percentile; 75th percentile). The nonparametric Mann–Whitney U test was used for quantitative variables. The sensitivity and specificity for Ultrasound were calculated using CT as the golden standard.

## 3. Results

### 3.1. iAAA Group: Case Series (Study 1)

In the iAAA group, the median age was 64 (61; 72) years, and all were of the male sex (*n* = 13, 100%). CRP and ESR levels were 34.5 (12.8; 66.3) mg/L and 38.5 (13.5; 114) mm/h, respectively. A substantial number (*n* = 8, 62%) of the patients were current smokers. Furthermore, the majority of the patients exhibited hypertension (*n* = 10, 77%), and hyperlipidemia (*n* = 9, 69%). Of these patients, 38.5% (*n* = 5) were diagnosed with (definite) IgG4-RD, four due to positive biopsies, and one probable IgG4-RD due to an IgG4 serum level higher than 2 g/L. The majority of patients reported symptoms commonly associated with (inflammatory) AAA, these being pain (62%, *n* = 8), loss of energy (54%, *n* = 7), and constipation (39%, *n* = 5), respectively. More than half of the patients (54%, *n* = 7) underwent surgical repair: four with open repair, and three by means of endovascular repair. Of these 13 patients, 11 (85%) received glucocorticoids for the treatment of iAAA. See Table 1.

All patients showed a typical hypoechogenic cuff surrounding the aortic wall eccentric to the calcified medial layer of the aneurysm (Figure 1B). The median maximum anteroposterior diameter of the aneurysm itself was 5.7 cm (4.8; 6.1). The median maximum measurement of the cuff was 6.4 mm (4.9; 8.4).

Of the 13 patients who were included in the case series, 11 already showed a cuff on the first available ultrasound, and two patients developed iAAA during the follow-up for a non-inflammatory AAA. The first patient exhibited an inflammatory cuff on ultrasound, 27 months after his first ultrasound for the evaluation of non-inflammatory AAA. The second patient exhibited an inflammatory cuff on ultrasound 6 months after the ultrasound for the non-inflammatory AAA follow-up (Figure 2).

### 3.2. AAA Group: A Feasibility Study (Study 2)

The AAA group, used to assess feasibility, consisted of 157 patients, with a median age of 74.5 (67; 80) years; 132 were male (84%). Of these 157 patients with an AAA and with both CT and Ultrasound imaging available, 147 (93.6%) patients showed no signs of a hypoechogenic cuff on ultrasound, as well as no sign of iAAA on CT imaging. Of the remaining ten patients, eight (5.1%) showed a cuff (mantle sign) surrounding the aortic wall on ultrasound imaging and showed a positive mantle sign on the CT, and were subsequently identified as having iAAA. In the remaining two patients (1.3%), an iAAA could not be ruled out based on ultrasound imaging alone, and subsequently were after a CT assessment considered it to be false positive (Figure 3). This results in a sensitivity of 100%, and a specificity of 98.7%.

Of these eight patients, who were identified as having iAAA, the median age was 65 years (63; 79), all were male (100%), and their CRP and ESR were at baseline 24 (9; 49) mg/L and 43 (11; 117) mm/h, respectively. The median follow-up was 3 years and 5 months (3 years 2 months; 4 years 5 months); one (12.5%) patient died during follow-up. One of the most serious complications of iAAA, hydronephrosis due to encasement of the ureters, was present in 50% (*n* = 4) of the cases, of which two patients (25%) had bilateral hydronephrosis. The majority of these patients were smokers as well, either current or former (*n* = 5, 83.3%). Furthermore, the majority of the patients were slightly overweight with a BMI of 25.6 kg/m^2^ (24.6; 29.5). Two of the patients had a confirmed (definite) IgG4-RD using the pathology of postoperatively acquired tissue; no probable IgG4-RD cases were present in this group. Of these patients, three (38%) reported abdominal pain, two (25%) reported fatigue, and two (25%) reported initially with a hydrocele. Seven (88%) were treated with corticosteroids, and four (50%) received surgical treatment for their (inflammatory) AAA (*n* = 2 open surgery, *n* = 2 endovascular repair). Of these eight patients the median maximum anteroposterior diameter of the aneurysm itself was 4.9 cm (4.3; 5.7). The median maximum measurement of the cuff was 4.3 mm (2.0; 8.4). A more complete description of the patient group is presented in Table 2. 

## 4. Discussion

With these data, we have demonstrated that inflammatory abdominal aortic aneurysms can be identified by using B-Mode ultrasound. An ultrasound image showing a hypoechogenic cuff surrounding the aortic wall, eccentric to the calcified medial layer of the aneurysm, is pathognomonic for iAAA. 

Furthermore, ultrasound is a standardized method to identify and follow up on individuals with AAA in standard care. Therefore, ultrasound has strong potential in identifying patients with AAA who have progressed to iAAA, and thus potentially are at risk for progression to the stage of retroperitoneal fibrosis, which can ultimately lead to hydronephrosis. 

Ultrasound is a valuable tool for diagnosing iAAA due to its relative portability, inexpensiveness, and lack of ionizing radiation, unlike other imaging tests such as CT or PET imaging. Ultrasound can provide detailed images of the aortic wall and—to a certain extent—surrounding tissues, allowing healthcare providers to accurately measure the size and shape of the aneurysm, and assess its growth over time. Furthermore, where some other modalities are either absent (e.g., PET-CT), or are not readily available, ultrasound imaging is more often available in hospitals, potentially preventing delays in the diagnosis of iAAA.

In our study, we first evaluated ultrasound imaging in cases with already proven iAAA in the case series group, and hence it was not feasible to evaluate false positive or negative cases in this instance. Hence, we subsequently trained a medical student to perform an initial evaluation for the presence of iAAA in a group of patients with a confirmed AAA diagnosis, which proved an effective way to rule out iAAA in a large part of the AAA group. Due to a low rule-out threshold, no patients were missed by either the student or the medical doctor and yet yielded a sensitivity of 100% and a specificity of 98.7%, which is higher than the sensitivity reported by the previous study on this topic, which was 60% [8]. This increase in sensitivity might be partially explained by the methodology of the previous study, which was limited in some aspects, such as clear inclusion and exclusion criteria for iAAA. However, it has to be mentioned that ultrasound imaging has made substantial improvements in the last decade, so these results might also be explained by the improvements in ultrasound imaging alone. These advances in ultrasound imaging are amongst other factors: real-time computer imaging which has increased the processing speed of ultrasound devices enabling the production of better images; additionally, regarding iAAA to a limited degree, improved transducer frequencies, as the frequencies needed for (inflammatory) AAA imaging have not changed considerably due to the scanning dept requirements. These improvements however are vital for other vascular inflammatory diseases, such as giant cell arteritis (GCA), and due to the scanning department can now utilize higher frequency transducers, and in turn produce higher resolution images, aiding in the diagnosis of GCA [12,13]. To do this they utilize an ultrasound phenomenon called a halo sign, which is a sign for vascular wall inflammation [14,15].

For iAAA cases with a positive ultrasound, it can be argued that a CT scan might still be warranted, owing to the inconclusive and potentially false positive ultrasound imaging in two of the ten ultrasounds flagged as having a cuff. In these cases, a CT scan will be able to give a better overview of the potential abnormal findings in the aorta, as well as outside of the aorta. 

Furthermore, the utilization of ultrasound imaging, which is already used in the follow-up of AAA, can aid in identifying iAAA earlier than is currently the case. This may possibly even result in the earlier treatment of iAAA and its underlying systemic diseases such as IgG4-RD, allowing interventional actions preventing the occurrence of RPF and the progression of the iAAA often resulting in hydronephrosis; this is supported by the findings in the iAAA case series (study 1), in which two patients developed a hypoechogenic cuff during ultrasound follow-up for their (non-inflammatory) AAA. However, due to the irregular shape of the cuff in some cases, the approach of cuff measurement may not be suitable for each patient and we believe more sophisticated methods for evaluating cuff area or even volume are essential.

Our data also show that the inflammation in AAA is most likely not the cause of AAA, as in two cases in our validation group the inflammation was picked up after the initial diagnosis of AAA. An iAAA is therefore most likely to be considered a consequence of AAA rather than the cause. This is a theory that was first published in the late 1980′s and is still held to be true to this day [16]. It is interesting to note however that the average age of patients with iAAA is 5–10 years younger than those with solely AAA, [2,9] which is something that was also observed in our AAA group (feasibility study). Reasons for this are not readily available but could be due to the fact that iAAA patients more often present themselves earlier with a raised ESR, a finding that was noticed in our groups as well. Another option is that they often present with physical symptoms as well, such as back, flank, or abdominal pain, [2] or that the inflammatory component of AAA had become unusually accentuated.

However, there are some limitations to our study as well. The design limited our ability to perform exact measurements of CT and US images, as these were performed retrospectively and with different scanners and scanning protocols; furthermore, ideally one would have made the ultrasound and CT imaging closer together in time, as the gap might have introduced a factor of uncertainty, due to the possible development of iAAA in between tests. Additionally, the limited number of patients in the iAAA group is a cause for possible bias. For example, there were in both groups no women observed with an iAAA diagnosis, possibly owing to the low amount of iAAA inclusions and the low risk of iAAA in the female sex. We would therefore recommend further research to be conducted in a prospective group of AAA patients using our data to set up a standardized assessment protocol and follow-up. This will also allow a standardized scanning procedure and measurement protocol.

## 5. Conclusions

This retrospective study indicates that iAAA might potentially be identified with ultrasound as a hypoechogenic cuff surrounding the aortic wall of the aneurysm. Ultrasound could be used to rule out iAAA safely. However, in positive ultrasound cases, additional CT imaging is still warranted. Ultrasound may facilitate earlier recognition and treatment of iAAA and underlying systemic inflammatory diseases such as IgG4-RD, allowing interventional actions preventing the occurrence of retroperitoneal fibrosis. Although this is the first study in 25 years to structurally investigate ultrasound in iAAA, our results need validation in a prospective study.

## Figures and Tables

**Figure 1 diagnostics-13-01669-f001:**
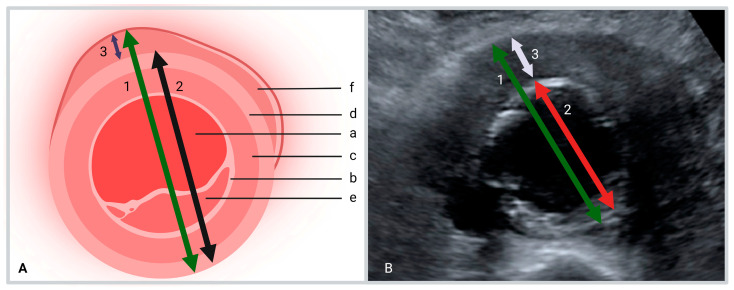
(**A**) (a) Lumen, (b) intima, (c) media, (d) adventitia, (e) thrombotic lesion, (f) inflammatory cuff. (**B**) (1) The maximum anteroposterior diameter of the aorta including the hypo-echogenic cuff. (2) The anteroposterior diameter of the aorta excluding the hypo-echogenic cuff. (3) The maximum measurement of the hypo-echogenic cuff.

**Figure 2 diagnostics-13-01669-f002:**
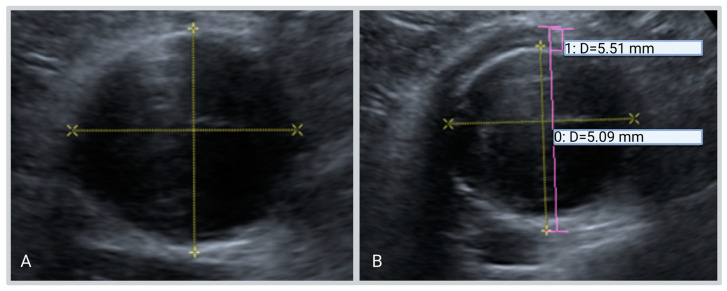
iAAA patient presenting with a new hypoechogenic cuff during follow-up: (**A**) Initial ultrasound; (**B**) ultrasound during follow-up.

**Figure 3 diagnostics-13-01669-f003:**
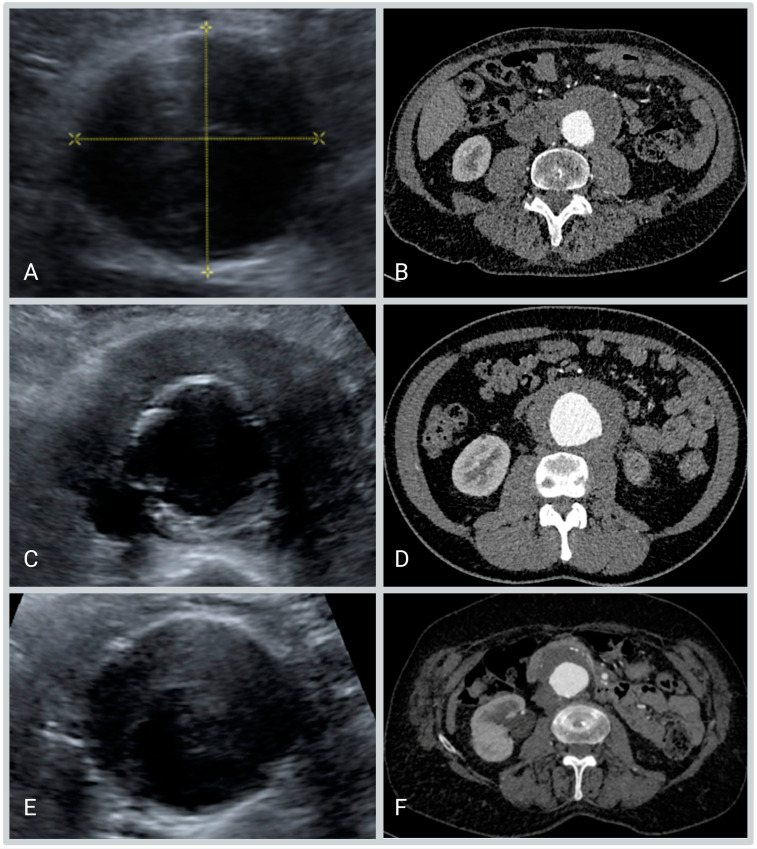
(**A**,**B**) Patient with no sign of iAAA on both Ultrasound and CT. (**C**,**D**) Patient with a cuff on ultrasound, as well as a mantle sign. (**E**,**F**) Patient with a false positive ultrasound, with the accompanying negative CT scan.

**Table 1 diagnostics-13-01669-t001:** Patient Characteristics of the iAAA Case Series.

	Patients (*n* = 13)
Median age, y (IQR)	64 (61; 72)
Sex, male, *n* (%)	13 (100%)
CRP; median (IQR), mg/L	35 (13; 66)
ESR; median (IQR), mm/h	39 (14; 114)
Current Smoking, *n* (%)	8 (62%)
Body Mass Index, median (IQR), kg/m^2^	26 (24; 29)
Hypertension, *n* (%)	10 (77%)
Hyperlipidemia, *n* (%)	9 (69%)
Type I or II DM, *n* (%)	0 (0%)
Cardiac disease, *n* (%)	4 (31%)
Pulmonary disease, *n* (%)	1 (8%)
Hydronephrosis, *n* (%)	2 (15%)
Definite IgG4-RD, *n* (%)	4 (31%)
Probable IgG4-RD, *n* (%)	1 (8%)
Other auto-immune disorder, *n* (%)	1 (8%)

**Table 2 diagnostics-13-01669-t002:** Patient characteristics of iAAA in the AAA group.

	Patients (*n* = 8)
Median age, y (IQR)	65 (63; 79)
Sex, male, *n* (%)	8 (100%)
CRP; median (IQR), mg/L (*n* = 7)	24 (9; 49)
ESR; median (IQR), mm/h (*n* = 4)	42.5 (11; 117)
Smoking, *n* (%) (*n* = 6)	5 (83.3%)
*Current smoking, n (%)*	1 (16.7%)
*Past smoking, n (%)*	4 (66.7%)
Body Mass Index, median (IQR), kg/m^2^ (*n* = 7)	25.6 (24.6; 29.5)
Hypertension, *n* (%) (*n* = 7)	4 (57.1%)
Type I or II DM, *n* (%)	1 (12.5%)
Cardiac disease, *n* (%) (*n* = 7)	1 (14.3%)
Pulmonary disease, *n* (%)	2 (25%)
Hydronephrosis, *n* (%)	4 (50%)
*Unilateral, n (%)*	2 (25%)
*Bilateral, n (%)*	2 (25%)
Definite IgG4-RD, *n* (%)	2 (25%)
*Other auto-immune disorder, n (%)*	0 (0%)

## Data Availability

For inquiries about the data used, please contact the corresponding author.

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
