# Peer review of "Ultrasound for the Detection of Inflammatory Abdominal Aortic Aneurysms: A Case and Validation Series"

_diagnostics, 2023, doi:10.3390/diagnostics13101669_

Round 1

Reviewer 1 Report

Interesting paper that carefully analyzes the ultrasound importance for the diagnosis and follow up of iAAA.

The ultrasound use (according to the method described) allows to reduce the indications to CT only if it is performed in a Centre with adequate experience.

The bibliography, which may seem poor, reports what is currently available in the literature except for only one recent paper (2022) which, however, does not appear important for this study (Multidisciplinary diagnosis and management of inflammatory aortic aneurysm. Jun Xu et al. J Vasc Surg 2022).

The usefulness of ultrasound increasing application in iAAA diagnosis and follow up allows cost reduction, easier access to the diagnosis, favouring a reduction in the use of X-rays and iodine contrast.

Author Response

Reply to reviewer 1:

We would like to thank the reviewer for his/her comments and corrections. Hereby, we reply to the comments.

  1. Comment: The bibliography, which may seem poor, reports what is currently available in the literature except for only one recent paper (2022) which, however, does not appear important for this study (Multidisciplinary diagnosis and management of inflammatory aortic aneurysm. Jun Xu et al. J Vasc Surg 2022).

Reply: We would like to thank the reviewer for the relevant point. We have considered adding this article in our reference section (cited as nr. 6), by citing it in a new, but small relevant section in our manuscript in the introduction (page: 2; lines: 51-52), the added text reads: “Having prompted some to classify iAAA as either IgG4-iAAA or non-IgG4-iAAA.”  

Reviewer 2 Report

This was a very interesting report and provided a novel opportunity to detect inflammatory AAAs during AAA periodic surveillance. We strongly suggested authors to address following issues to substantially improve study quality. 

1. As a Diagnostics journal, strongly recommend authors to provide individual ultrasound images for 13 patients in case series and 8 patients found in feasibility studies. Also, providing statistical data on parameters mentioned in Figure 1 would be a big plus. 

2. Did authors follow up 8 patients identified in feasibility study (even only medical record data are fine)? If yes, please provide the data on follow up period and outcomes.

3. In case series, is possible for authors to get information on how many patients were asymptomatic or symptomatic from the medical records?

4. Because there were only 13 patients in case series, please address this as a limitation particularly possible false positive and false negative.  

5. Please provide the references or evidence that IgG4 > 2 g/L is associated with IgG4-RD.

Author Response

Reply to reviewer 2:

We appreciate the reviewer's comments and suggestions. We hereby reply to them:

  1. Comment: As a Diagnostics journal, strongly recommend authors to provide individual ultrasound images for 13 patients in case series and 8 patients found in feasibility studies. Also, providing statistical data on parameters mentioned in Figure 1 would be a big plus. 

Reply: We appreciate the reviewer's suggestions, and have taken them to heart. We have added the individual ultrasound imaging stills from both the case series, and feasibility study in the Supplementary. Concerning the last part of this remark, for the case series, statistical data was already present in the manuscript (page 5; lines 192-193). However, we initially didn’t add this for the feasibility group, but have now added it accordingly (page 7; lines 231-233). Reading: “Of these 8 patients the median maximum anteroposterior diameter of the aneurysm itself was 4.9 cm (4.3;5.7). The median maximum measurement of the cuff was 4.3 mm (2.0;8.4).” By adding this however, we felt an additional statement was needed in the discussion, which was added on page 8 (lines 288-291) Reading: However, due to the irregular shape of the cuff in some cases, the approach of cuff measurement may not be suitable for each patient and we believe more sophisticated methods for evaluating cuff area or even volume are essential.”

  1. Comment: Did authors follow up 8 patients identified in feasibility study (even only medical record data are fine)? If yes, please provide the data on follow up period and outcomes.

Reply:  We would like to thank the reviewer for this question, and accordingly added the relevant data available to our manuscript in the results section starting by adding “The median follow-up was 3 years and 5 months (3 years 2 months;4 years 5months), 1 1 (12.5%) patient died during follow-up.” (page 6; lines 220-221). And by subsequently adding this additional part, reading the following: “Of these patients, 3 (38%) reported abdominal pain, 2 (25%) reported fatigue, and 2 (25%) reported initially with a hydrocele. 7 (88%) were treated with corticosteroids, and 4 (50%) received surgical treatment for their (inflammatory) AAA (n=2 open surgery, n=2 endovascular repair).” (pages 6-7; lines 227-233).

  1. Comment: In case series, is possible for authors to get information on how many patients were asymptomatic or symptomatic from the medical records?

Reply: We appreciate the reviewer’s question, and added similarly to comment 2, available data on presenting symptoms, as well as following treatment regimen. We added this to results section on page 4 (lines 184-188).  Now reading: “The majority of patients reported symptoms commonly associated with (inflammatory) AAA. Being pain (62%, n= 8), loss of energy (54%, n=7), and constipation (39%, n=5) respectively. More than half of the patients (54%, n=7) patients underwent surgical repair, of which 4 with open repair, and 3 by means of endovascular repair. Of these 13 patients, 11 (85%) received glucocorticoids for the treatment of iAAA.”

  1. Comment: Because there were only 13 patients in case series, please address this as a limitation particularly possible false positive and false negative. 

Reply: We appreciate the reviewer’s remark, and added clarification on our inclusion methods for the initial case series in the methods section of the manuscript. Owing to the fact that all these patients already had a confirmed diagnosis made with CT imaging. (page 2; lines 86-87). Now reading: “In the first study, a case series was retrospectively assembled, consisting of 13 patients who visited our hospital between 2013 and 2020, without the prior surgical intervention of the abdominal aorta and who were diagnosed with iAAA according to the guidelines of the European Society of Vascular Surgery (ESVS), had CT imaging available.” The part in bold is the added part.

However, we did also try to clarify this in the discussion as well (page 7; lines 253-257) which now reads after some minor restructuring of the paragraph, as the following: “In our study, we first evaluated ultrasound imaging in cases with already proven iAAA in the case series group, and hence it was not feasible to evaluate false positive or negative cases in this instance. Hence, we subsequently trained a medical student to do an initial evaluation for the presence of iAAA, in a group of patients with a confirmed AAA diagnosis, which proved an effective way to rule out iAAA in a large part of the AAA group.”

  1. Comment: Please provide the references or evidence that IgG4 > 2 g/L is associated with IgG4-RD.

Reply: We appreciate the reviewer’s feedback in this regard, and agreeing with this comment we chose to add the following reference (found in the reference section as nr. 11).

  1. Hao, M. Liu, G. Fan, X. Yang, and J. Li, “Diagnostic value of serum IgG4 for IgG4-related disease,” Medicine (United States), vol. 95, no. 21. Lippincott Williams and Wilkins, May 01, 2016. doi: 10.1097/MD.0000000000003785.

Which is a PRISMA compliant systematic review and meta-analysis, which gives >135mg/dL (1.35g/L) as an elevated IgG4. Our university laboratory has its own reference range which equates to >2 g/L.

Kind regards,

Berend Slijkhuis

On behalf of all coauthors.

University Medical Centre Groningen, Internal Medicine, Division Vascular Medicine, Internal code AA41 Hanzeplein 1, 9700 RB, Groningen, The Netherlands.

Telephone number: +31 6 244 922 13

E-mail address: [email protected]

Round 2

Reviewer 2 Report

Thank you for responding to my concerns.